# 🐤 Cappy: Outperforming and Boosting Large Multi-Task LMs with a Small Scorer

**Bowen Tan**[1],[*] **Yun Zhu**[2], **Lijuan Liu**[2], **Eric Xing**[1,3,5], **Zhiting Hu**[4], **Jindong Chen**[2]
[1]Carnegie Mellon University,   [2]Google Research,   [3]Petuum Inc.,   [4]UC San Diego,
[5]Mohamed bin Zayed University of Artificial Intelligence
{btan2, epxing}@andrew.cmu.edu, zhh019@ucsd.edu,
{yunzhu, lijuanliu, jdchen}@google.com

## Abstract

Large language models (LLMs) such as T0, FLAN, and OPT-IML, excel in multi-tasking under a unified instruction-following paradigm, where they also exhibit remarkable generalization abilities to unseen tasks. Despite their impressive performance, these LLMs, with sizes ranging from several billion to hundreds of billions of parameters, demand substantial computational resources, making their training and inference expensive and inefficient. Furthermore, adapting these models to downstream applications, particularly complex tasks, is often unfeasible due to the extensive hardware requirements for finetuning, even when utilizing parameter-efficient approaches such as prompt tuning. Additionally, the most powerful multi-task LLMs, such as OPT-IML-175B and FLAN-PaLM-540B, are not publicly accessible, severely limiting their customization potential. To address these challenges, we introduce a pretrained small scorer, *Cappy*, designed to enhance the performance and efficiency of multi-task LLMs. With merely 360 million parameters, Cappy functions either independently on classification tasks or serve as an auxiliary component for LLMs, boosting their performance. Moreover, Cappy enables efficiently integrating downstream supervision without requiring LLM finetuning nor the access to their parameters. Our experiments demonstrate that, when working independently on 11 language understanding tasks from PromptSource, Cappy outperforms LLMs that are several orders of magnitude larger. Besides, on 45 complex tasks from BIG-Bench, Cappy boosts the performance of the advanced multi-task LLM, FLAN-T5, by a large margin. Furthermore, Cappy is flexible to cooperate with other LLM adaptations, including finetuning and in-context learning, offering additional performance enhancement. [2]

## 1 Introduction

Large language models (LLMs) have led to a new paradigm that seeks to unify various natural language processing (NLP) tasks within an instruction-following framework. This paradigm is exemplified by the recent multi-task LLMs, such as T0 [23], FLAN [30, 4], and OPT-IML [10]. These models are trained with data from many tasks: for each task, following a task-specific template, each labeled example is converted into an instruction (e.g., `"Put the concepts together to form a sentence: ski, mountain, skier."`) and a corresponding response (e.g., `"Skier skis down the mountain"`). Such `(instruction, response)` pairs are then used to train the LLM, resulting in a conditional generation model that takes input of a data example as an instruction

---

[*]Work done during an internship at Google.

[2]Code and model available at `https://github.com/tanyuqian/cappy` and `https://huggingface.co/btan2/cappy-large`, respectively.

37th Conference on Neural Information Processing Systems (NeurIPS 2023).

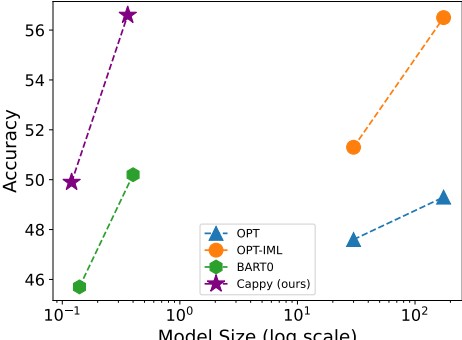

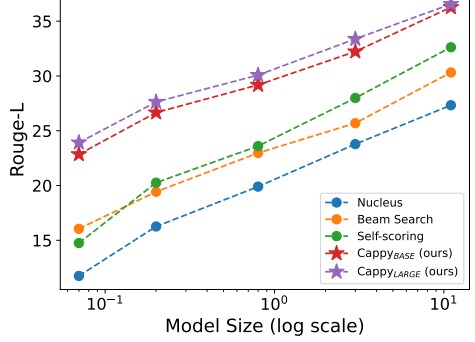

Figure 1: Cappy *outperforms* multi-task LLMs: The overall accuracy averaged over 11 test tasks from PromptSource. Every dashed line connects different sizes of the same model. Lines positioned more towards the *upper left* denote models that are more efficient and yield superior performance.

Figure 2: Cappy *boosts* multi-task LLMs: The averaged Rouge-L score over 45 complex tasks within BIG-Bench. Every dashed line represents an approach working on LLMs of various sizes. Self-scoring refers to using the cross-entropy of LLM to select responses.

and generates a response. Moreover, these multi-task LLMs have exhibited remarkable task-wise generalization capabilities. That is, they can address unseen tasks by understanding and solving brand-new instructions.

Due to the complexity of understanding and resolving various tasks solely via instructions, the sizes of these multi-task LLMs typically span from several billion parameters to hundreds of billions, such as T0-11B [23] and OPT-IML-175B [10]. As a result, operating such sizable models poses significant challenges to the majority of LLM users, because they demand considerable computational power and impose substantial requirements on the memory capacities of GPUs/TPUs, making their training and inference expensive and inefficient.

In practical applications, harnessing a single multi-task LLM to manage all conceivable tasks in a zero-shot manner remains challenging, particularly when dealing with complex tasks, personalized tasks and those that cannot be succinctly defined using instructions. On the other hand, the size of downstream training data is usually insufficient to well train a model without incorporating rich prior knowledge. Hence, it is long desired to adapt LLMs with downstream supervision. Yet, the adaptation process presents three significant obstacles: the extensive storage to maintain a unique LLM copy for each downstream task; the considerable memory demands on GPUs/TPUs; and the unavailability of the most powerful multi-task LLMs, such as OPT-IML-175B [10] and FLAN-PaLM-540B [4]. Certain parameter-efficient tuning strategies, including prompt tuning [18] and adapters [8], substantially diminish storage requirements, but they still perform back-propagation through the LLM parameters during the tuning process, thereby their memory demands keep high. Additionally, some in-context learning techniques [5] circumvent parameter tuning by integrating a limited number of supervised examples into the instruction. However, these techniques are constrained by the model's maximum input length, which permits only a few samples to guide task resolution.

In this work, we propose a novel approach to enhance the performance and efficiency of multi-task LLMs. Specifically, we introduce a lightweight pretrained scorer, *Cappy*, based on a continual pretraining on top of RoBERTa [20], with merely 360 million parameters. Cappy takes in an instruction and a candidate response as input, and produces a score between 0 and 1, indicating an estimated correctness of the response with respect to the instruction. Naturally, we formulate Cappy's pretraining as a regression problem. This anticipates training data in the form of (`instruction`, `response`) pairs that correspond to various correctness score annotations. To generate the desired training data from multiple pretrain datasets that solely contain instructions and their ground truth responses, we propose a weakly-supervised approach with data augmentation through the use of existing multi-task LLMs. As a result, we obtain a large and effective regression pretraining dataset with diverse correctness score annotations ranging from 0 to 1.

To apply Cappy to practical problem-solving scenarios, we suggest an intuitive approach in an candidate selection style. Specifically, Cappy works independently on classification tasks by selecting

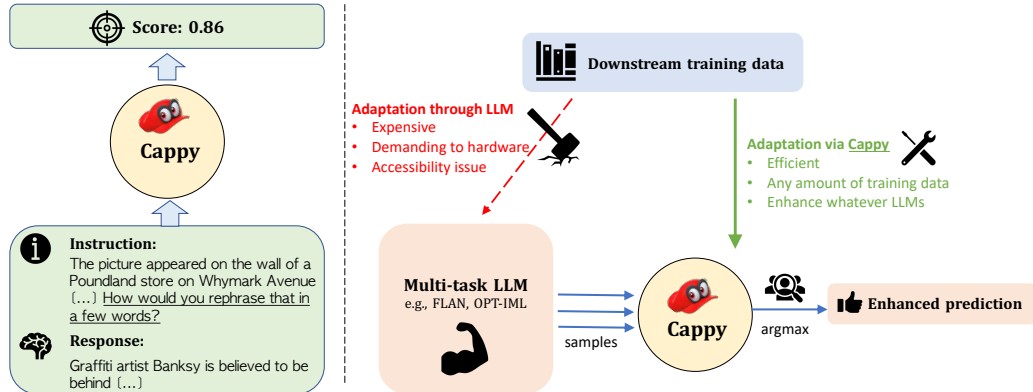

Figure 3: **(left)** The modeling of Cappy. **(right)** Illustration of Cappy's application in enhancing multi-task LLMs, and the comparison between downstream adaptation through Cappy and approaches that rely on LLM's parameters, such as finetuning and prompt tuning.

the answer choice that produces the highest score. Furthermore, beyond the standalone use, Cappy serves as an auxiliary component of existing multi-task LLMs, choosing the most appropriate output from a set of candidates generated by the LLM. In this case, Cappy allows for effective and efficient adaptation to complex tasks through the incorporation of downstream supervision, without requiring finetuning the multi-task LLM or the access to its parameters. Remarkably, Cappy exhibits flexibility in collaboration with other LLM adaptations, such as finetuning and in-context learning.

We validate Cappy through an extensive suite of held-out tasks distinct from those incorporated in its pretraining. The overall performance is as shown in Fig. 1 and Fig. 2. Specifically, on 11 language understanding tasks drawn from PromptSource [1], Cappy, with 360 million parameters, outperforms OPT-IML-30B and OPT-175B significantly, and matches the best ones among previous multi-task LLMs. Besides, on 45 diverse complex tasks from BIG-Bench [25], Cappy consistently boosts the performance of the advanced multi-task LLM, FLAN-T5, by a large margin. Furthermore, Cappy offers additional performance enhancement when applied together with finetuning or in-context learning. Our subsequent ablation study proves the significance of our proposed pretraining and data augmentation strategies.

## 2 Related Work

**LLMs for Instruction Following and Multi-task Prompted Training** The scaling up of language models brings them increasingly strong capabilities, culminating in a general paradigm of addressing diverse problems in a unified instruction-following manner. There are two primary approaches of such LLMs, each distinguished by the purpose of their instructions. The first approach emphasizes compliance with arbitrary human instructions, often in a question-and-answer or dialogue format (e.g., `"I have to make a difficult decision.  What should I do?"`). Models such as GPT-4 [21] and Vicuna [3] are designed to respond to these instructions with the goal of maximizing user satisfaction. These models are typically trained through Reinforcement Learning with Human Feedback (RLHF) [13], leveraging extensive human annotations. Their quantitative evaluation also heavily depends on human judgment [33]. The second approach, however, is primarily devoted to resolving well-defined NLP tasks. In this context, each data instance adheres to a task-specific template, and is transformed into an instruction (e.g., `"Put the concepts together to form a sentence:  ski, mountain, skier."`) and a corresponding response (e.g., `"A skier skis down the mountain."`). Multi-task LLMs, such as OPT-IML [10], FLAN [30, 4], and T0 [23], are pretrained via multi-task prompted training. This process trains models as a unified conditional generation task using pairs of instructions and responses from multiple upstream pretraining tasks. These models are typically assessed based on performance on held-out test tasks, utilizing traditional evaluation metrics such as accuracy, Rouge scores [17], and so forth. In this study, our primary focus is on the second approach, i.e., multi-task LLMs, given its more straightforward evaluation. Nonetheless, we posit that there is no significant obstacle to apply our proposed methodologies to more humanish instructions, which we leave as our future direction.

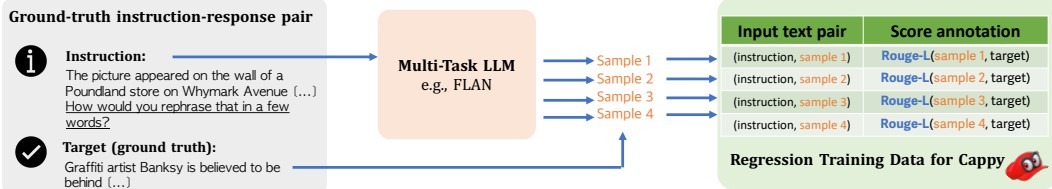

Figure 4: Data augmentation with a multi-task LLM to construct weakly supervised regression dataset for Cappy's pretraining and finetuning, as described in Sec. 3.2.

**Adaptation of LLMs** The size of LLMs makes their finetuning for downstream tasks particularly challenging, primarily due to three issues. Firstly, finetuning necessitates the creation of a new copy of an LLM for each specific downstream task. This is unacceptable for many applications. Secondly, fine-tuning an LLM demands considerable device memory because of the back-propagation through the LLMs, which is achievable only with high-end GPU/TPU clusters. Thirdly, the most powerful LLMs, such as FLAN-PaLM-540B [4] and GPT-4 [21], are closed-source and thus inaccessible for fine-tuning. A collection of parameter-efficient LLM adaptation techniques, including prompt tuning [18] and adapters [8], like prefix tuning [15] and LoRA [7], have largely mitigated the storage issue by decreasing the number of tunable parameters. However, these methods still require back propagation through the original LLM weights to update the prompt or the adapter, leaving the second and third issues remain significant barriers in LLM adaptation. Certain in-context learning techniques [5] circumvent LLM's parameter tuning by appending training examples to the instruction. However, the instruction length is limited by the model's maximum input length, such in-context learning techniques allow for only a finite number of samples to guide the task-solving process. In this work, we propose the adaptation of multi-task LLMs by employing Cappy to incorporate downstream supervision. This approach enables any number of training examples without necessitating LLM finetuning or access to its parameters. Hence, the LLM serves merely as a black box, and Cappy is even compatible with WebAPIs of LLMs. Importantly, Cappy can also be deployed in conjunction with other LLM adaptations, such as finetuning and in-context learning.

**Ranking-based Models** Ranking is a pivotal component of information retrieval systems, notably in search engines and recommendation systems [19]. It involves sorting vast numbers of documents to find content pertinent to a specific query. Recently, ranking has been adapted for NLP tasks to aggregate answers [14], and cater to human preferences [2, 6]. Furthermore, in the emergent domain of reinforcement learning from human feedback (RLHF) [26], ranking models trained on human-ranked model outputs serve as reward providers for the training RL agents. Concurrently with this work, [32] proposes a unified ranking model solving information alignment style tasks, such as natural language inference and paraphrase detection. In this work, Cappy is conceptually a ranking model for multi-task learning. Unlike methodologies specifically designed for question answering [12] or summarization [22], Cappy offers extensive generalizability across multi-task scenarios. Additionally, in contrast to RLHF reward models, Cappy doesn't rely on expensive human-annotated data, enabling large-scale pretraining.

## 3 The Cappy Scorer

### 3.1 Modeling

Cappy adopts the architecture of RoBERTa [20] with a linear layer on the top as a regression head. The input of Cappy is a pair of text, comprising an instruction and a response, and the output is a scalar score ranging from 0 to 1. This score indicates an estimation of the correctness of the response with regard to the task instance described in the instruction.

### 3.2 Pretraining

Cappy's pretraining uses the same dataset collection that is utilized by T0 [23]. This collection is comprised of 39 diverse datasets from PromptSource [1], encompassing a wide range of task types, such as question answering, sentiment analysis, and summarization, etc. Each dataset is associated with one or multiple templates, converting each instance from the original datasets into a instruction

paired with its ground truth response. Following the pretraining configuration of T0, the size of each dataset is limited to a maximum of 500,000 examples.

In light of Cappy's regression modeling, each data instance during pretraining must have a text pair `(instruction, response)`, coupled with an correctness annotation for the response in relation to the instruction. A diverse array of score annotations is a pivotal aspect of a regression dataset. However, the text pairs in our pretraining datasets merely contain instructions with their ground truth responses, hence each text pair invariably has a correctness score of 1.0. This could culminate in a critical lack of label diversity throughout Cappy's pretraining. To address this issue, we propose a data construction approach to produce Cappy's pretraining dataset with correctness annotations that diversely range from 0 to 1. The data construction consists of three components:

**Ground Truth (score 1.0)** This component encompasses all ground truth instruction-response pairs from the pretraining dataset. Each pair is assigned a correctness annotation of 1.0.

**Incorrect Responses (score 0.0)** We engineer incorrect data points by creating mismatched instruction-response pairs from the original datasets. For classification datasets, each instruction is paired with all incorrect answer choices. For generation datasets, each instruction is arbitrarily paired with a ground truth response from a distinct data point within the dataset.

**Data Augmentation (score within** $[0, 1]$**)** In addition to purely correct or incorrect data samples, we fabricate instruction-response pairs with scores ranging between 0 and 1. This is achieved through data augmentation applied across all generation task instances. For every instance within a generation task, we leverage an existing multi-task LLM to generate multiple responses by sampling conditioned on the given instruction. Subsequently, we assign an annotation to the pair formed by the instruction and every response, using the similarity between the response and the ground truth response of the instance. Specifically, we employ Rouge-L [17] to calculate this similarity as a form of weak supervision. as it has been widely recognized as a reliable metric for overall performance in multi-task environments and has demonstrated a strong alignment with human evaluation [29]. In practice, our augmented samples are generated by two multi-task LLMs, BART0 [16] and T0-3B [23]. For each instance within a generation task, both these models generate two samples using the top-k and top-p sampling, respectively.

Consequently, we collect a pretraining dataset comprised of 160 million instances, each in the format of `(instruction, response, score)`. Cappy is initialized as RoBERTa and optimized using the AdamW optimizer with an L2 regression loss. The optimization process involves a learning rate of $10^{-6}$, a warmup rate of 0.1, and an effective batch size of 1024. In alignment with the RoBERTa variants, Cappy is also offered in two distinct versions: the smaller $\text{Cappy}_{\text{BASE}}$(120M parameters), and the $\text{Cappy}_{\text{LARGE}}$(360M parameters).

### 3.3 Applying Cappy

Cappy solves practical tasks within a candidate-selection mechanism. More specifically, given an instruction and a set of candidate responses, Cappy produces a score for each candidate. This is achieved by inputting the instruction alongside each individual response, and then assigning the response with the highest score as its prediction. In classification tasks, all candidate responses are inherently predefined. For example, the options are `{positive, negative}` in a sentiment classification task. In such scenarios, Cappy functions independently. On the other hand, in generation tasks, candidate responses are not pre-defined, requiring an existing multi-task LLM to yield the candidate responses. In this case, Cappy serves as an auxiliary component of the multi-task LLM, enhancing its decoding.

### 3.4 Adapting Multi-task LLMs

When there is available downstream training data, Cappy enables effective and efficient adaptation of multi-task LLMs on downstream tasks. Specifically, we propose to integrate downstream task information into LLM's predictions through the finetuning of Cappy. To elaborate, a downstream regression dataset can be acquired through a data annotation process same as the approach utilized during the pretraining data construction (§3.2). Then, Cappy can be finetuned on this regression dataset. As a result, the finetuned Cappy collaborates with a multi-task LLM, boosting the LLM's performance on the downstream task.

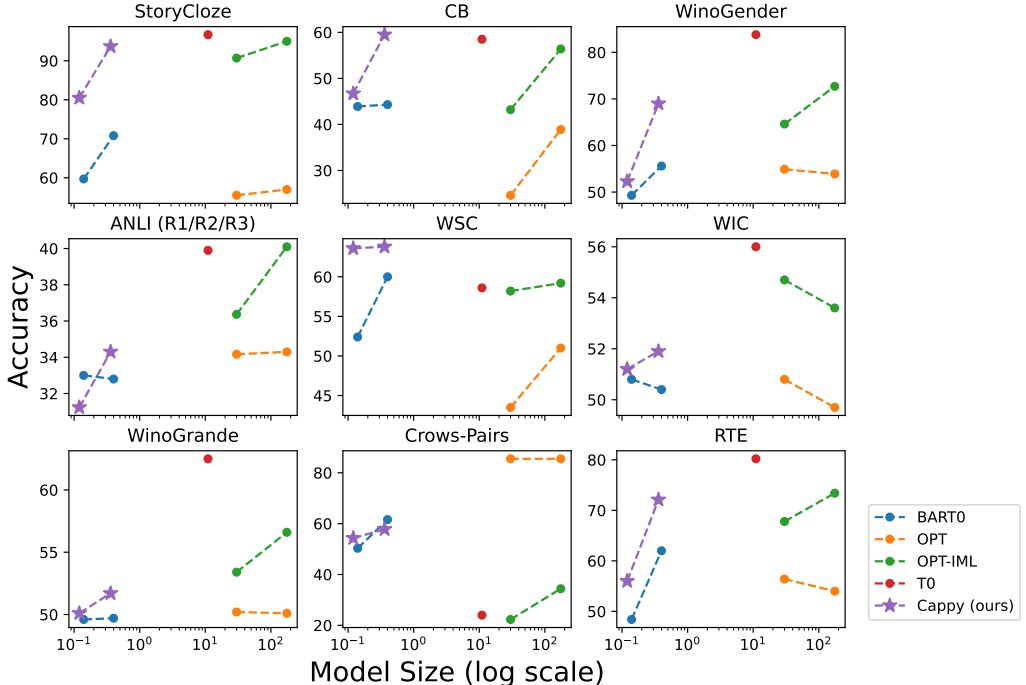

Figure 5: The performance of Cappy and multi-task LLMs on various test datasets. A series of dashed lines are used to connect different sizes of the same model, such as OPT-30B and OPT-175B. Lines or points positioned more towards the *upper left* of the diagram denote models that are more efficient and yield superior performance. Each diagram corresponds to a specific test task, with the exception of ANLI that represents three different tasks (ANLI-R1/R2/R3).

In contrast to other LLM tuning strategies such as finetuning and prompt tuning, adapting LLMs with Cappy on downstream tasks avoids the need for back-propagation through LLM parameters. Therefore, it significantly reduces the high demand for device memory. Besides, the adaptation with Cappy does not rely on the access to the LLM parameters, making it compatable with closed-source multi-task LLMs, such as the ones only accessible via WebAPIs. Compared with in-context learning approaches which circumvent model tuning by attaching training examples to the instruction prefix, Cappy is not restricted by the LLM's maximum input length. Thus, Cappy can incorporate an unlimited number of downstream training examples. Moreover, Cappy is flexible to work together with other adaptation methods, such as finetuning and in-context learning, further boosting their overall performance, as we demonstrate in experiments.

## 4   Experiments

All the experiments of this work, including the pretraining of Cappy and downstream adaptations, are conducted on Google TPU-v4 [11], and all the code is implemented with Redco [27], a lightweight toolkit for automating distributed training.

### 4.1   Zero-shot Performance on PromptSource

Our evaluation aligns with the ones used by OPT-IML and T0. We assess performance on 11 held-out language understanding tasks in PromptSource [1], all of which are in classification style. These tasks are categorically distinct from those utilized in the pretraining datasets. Our baselines include multi-task LLMs, i.e., OPT, OPT-IML, T0, and BART0, and a RLHF reward model trained with human feedback data, released by LAION-AI [3]. Following the answer selection strategy employed

---

[3]`https://huggingface.co/OpenAssistant/reward-model-deberta-v3-large-v2`

| Model | Accuracy |
|---|---|
| BART0$_{BASE}$-140M | 45.7 |
| BART0$_{LARGE}$-400M | 50.2 |
| RLHF-RM-185M | 43.6 |
| RLHF-RM-435M | 53.3 |
| OPT-30B | 47.6 |
| OPT-IML-30B | 51.3 |
| OPT-175B | 49.3 |
| OPT-IML-175B | 56.5 |
| T0-11B | 58.2 |
| Cappy$_{BASE}$-120M | 49.9 |
| Cappy$_{LARGE}$-360M | 56.6 |

Table 1: The overall accuracy averaged over 11 held-out test tasks from PromptSource in a zero-shot setting. "RLHF-RM" refers to the RLHF reward model mentioned in Section 4.1.

| | *Frozen* FLAN-T5 | | | | |
|---|---|---|---|---|---|
| | Small | Base | Large | XL | XXL |
| Sampling | 11.43 | 15.79 | 19.62 | 23.22 | 25.73 |
| Temperature | 12.01 | 17.06 | 20.05 | 24.27 | 27.10 |
| Top-K | 11.52 | 15.75 | 19.76 | 22.67 | 25.82 |
| Nucleus | 11.92 | 16.62 | 20.20 | 24.17 | 26.90 |
| Beam Search | 16.40 | 19.86 | 23.48 | 26.12 | 29.66 |
| Self-scoring | 15.08 | 20.71 | 24.12 | 28.47 | 32.02 |
| Cappy$_{BASE}$ | 23.36 | 27.26 | 29.83 | 32.79 | 36.63 |
| Cappy$_{LARGE}$ | **24.45** | **28.25** | **30.75** | **33.97** | **36.93** |
| ICL + Nucleus | 16.37 | 20.46 | 23.65 | 28.64 | 32.70 |
| ICL + Self-scoring | 20.61 | 24.42 | 27.00 | 32.56 | 36.37 |
| ICL + Cappy$_{LARGE}$ | **26.18** | **28.65** | **31.84** | **36.41** | **38.48** |

Table 2: The averaged Rouge-L score over 45 BIG-Bench tasks. The backbone FLAN-T5 models are frozen. "ICL" refers to in-context learning, i.e., attaching training examples in the prefix of instruction. We include more prompt-tuning related results in the appendix.

by T0 and OPT-IML, predictions from these multi-task LLMs are determined by the answer choice that yields the highest model likelihood. FLAN is not considered among the baselines, as the the test tasks are included in its pretraining tasks. We calculate the performance for each task by averaging the results across all associated prompts.

The outcomes of the 11 tasks are presented in Figure 5, with the averaged overall performance of each model summarized in Table 1. From the per-task figures, Cappy consistently outperforms BART0 models, which are comparable in size to Cappy, and also surpasses OPT and OPT-IML in the majority of tasks. From the overall accurary in Table 1, we can summarize that our Cappy$_{BASE}$model yields performance comparable to that of OPT-30B, OPT-IML-30B, and OPT-175B. Furthermore, our larger 360M Cappy$_{LARGE}$model performs at a level consistent with T0-11B and OPT-IML-175B. These findings highlight Cappy's superior performance and parameter efficiency in comparison to existing multi-task LLMs. This improved performance can be credited to Cappy's scoring-based pretraining strategy, which integrates contrastive information by differentiating between high-quality and low-quality responses. On the contrary, previous multi-task LLMs depend exclusively on teacher-forcing training that utilizes only the ground truth responses.

## 4.2 Adaptation to BIG-Bench

We examine the adaptation of multi-task LLMs with Cappy on complex tasks from BIG-Bench [25], a set of manually curated, challenging tasks that are considered beyond the capability of many LLMs. We focus on all the 45 generation tasks within BIG-Bench, specifically those that do not offer pre-established answer choices. The train/test split is provided by TaskSource [24]. The training sizes of these tasks are variable, with a minimum of 14 and a maximum of 50,000 instances. The median training size is 876. For each task, we finetune Cappy with an AdamW optimizer for 400 steps with a learning rate of $2 \times 10^{-5}$ and an effective batch size of 256. We evaluate the performance using the Rouge-L score on every test set, reporting the average score across 45 tests. In this experiment, all variants of FLAN-T5 serve as the backbone LLMs.

We incorporate these approaches in comparison, including: **Sampling**: Standard token-by-token sampling; **Temperature**: Sampling every token with a distribution temperature of 0.9; **Top-K**: Top-k sampling with k=40; **Nucleus**: Nucleus sampling with top-p=0.95; **Beam Search**: Beam search with a width of 4; **Self-scoring**: We collect four generated samples using all the sampling-based decoding strategies above, plus the top sample from beam search [4], in total $4 \times 4 + 1 = 17$ samples. With all the samples, self-scoring selects the best one as prediction based on the model log-likelihood; **Cappy**: conducting sample selection on the same set of samples as in self-scoring, but based on Cappy's

---
[4]We don't collect multiple samples by beam search, because the Jax API for beam search in huggingface-transformers only returns the top sample.

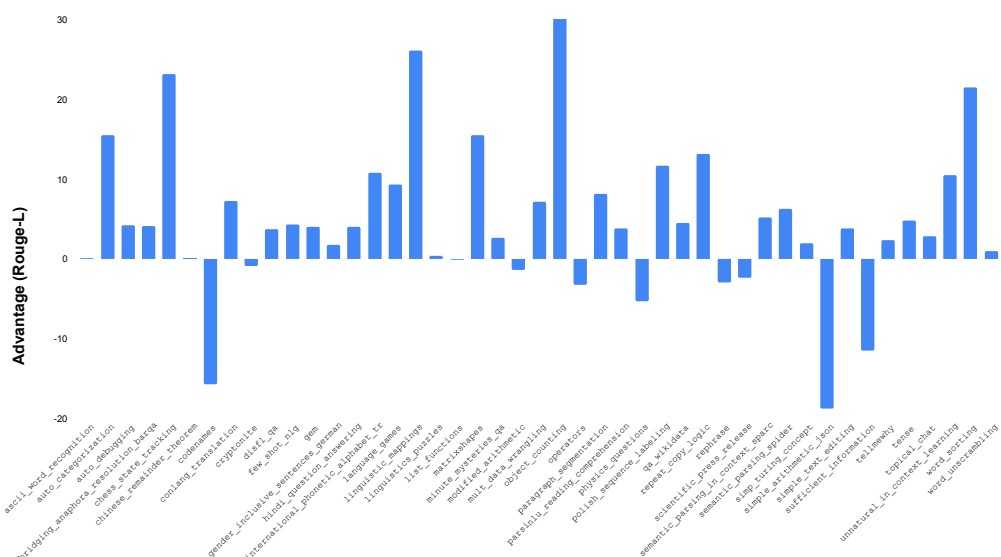

Figure 6: Advantage of Cappy scoring over FLAN-T5-XXL's self-scoring, on 45 BIG-Bench tasks. The x-axis is the names of the tasks.

scoring. Besides, we include a setting of *in-context learning (ICL)* by attaching training examples at the beginning of the instruction until the model's sequence length limit is reached. [5]

The foundational FLAN-T5 models are frozen, that is, not finetuned. The results are displayed in Table 2. They suggests that Cappy enhances the performance of FLAN-T5 models by a large margin, consistently outperforming the most effective baseline achieved through sample selection using self-scoring of the LLM itself.

As mentioned in Section 3.4, Cappy exhibits flexibility by enabling collaboration with other LLM adaptations. The performance of Cappy working together with finetuning and in-context learning, is presented in Table 2 and Table 3, respectively. The result demonstrate that Cappy keeps boosting performance on top of other adaptations. This can be attributed to the unique downstream knowledge that Cappy acquires from downstream training data. More precisely, while other LLM adaptations predominantly depend on traditional ground truth instruction-response pairs for learning, Cappy extracts and harnesses the contrastive information by its regression training data constructed with our proposed method.

|  | *Finetuned* FLAN-T5 | | |
|  | Small | Base | Large |
|---|---|---|---|
| Sampling | 29.34 | 37.93 | 43.45 |
| Temperature | 29.83 | 38.21 | 43.65 |
| Top-K | 30.06 | 37.48 | 43.86 |
| Nucleus | 30.12 | 37.87 | 44.35 |
| Beam Search | 32.00 | 39.21 | 44.52 |
| Self-scoring | 33.95 | 41.00 | 46.49 |
| Cappy$_{BASE}$ | 37.79 | 43.73 | 47.22 |
| Cappy$_{LARGE}$ | **39.74** | **45.18** | **48.98** |

Table 3: The averaged Rouge-L score over BIG-Bench tasks, when the backbone FLAN-T5 models are also finetuned.

### 4.2.1 Analysis on Cappy's Scoring

To further understand Cappy's scoring, we conducted a task-by-task analysis for all the 45 BIG-Bench tasks, comparing the performance between Cappy's scoring with the self-scoring of the multi-task LLM - FLAN-T5-XXL (11B). Figure 6 displays the performance advantage of Cappy over FLAN-T5-XXL's self-scoring (with negative values indicating Cappy's disadvantage). The results reveal that, for most tasks, Cappy consistently maintains a Rouge-L approximately 5 points higher than the LLM's self-scoring. However, there are 3 tasks on which Cappy exhibits a significant disadvantage: `codenames`, `simple_arithmetic_json`, and `sufficient_information`.

We showcase examples of these 3 tasks in Table 4. Upon examining the `codenames` examples, we find that the instructions often contain disjointed words with diverse meanings without syntactic

---

[5]We also conduct prompt tuning for the adaptation of Big-Bench. Results are discussed in the Appendix.

| **Task name:** `codenames` | | |
| --- | --- | --- |

**Instruction:** Try to identify the 3 words best associated with the word PAJAMAS from the following list: nude, judge, sleep, einstein, groom, troll, wish, sun, quarter, halloween, brain, stamp, wedding, slipper, minotaur, pad, tip, crusader, helmet. Give your answer in alphabetical order.
**Target:** nude, sleep, slipper

**Instruction:** Try to identify the 1 word best associated with the word PREHISTORIC from the following list: boom, new york, cotton, green, ball, pumpkin, force, suit, board, jet, mug, head, mammoth, seal, day, engine. Give your answer in alphabetical order.
**Target:** mammoth

| **Task name:** `simple_arithmetic_json` | | |
| --- | --- | --- |

**Instruction:** 5 + 0 =
**Target:** 5

**Instruction:** 348 + 227 =
**Target:** 575

| **Task name:** `sufficient_information` | | |
| --- | --- | --- |

**Instruction:** Ed, Jeff, E-Jay, and Michael are in a circle. Ed is on Jeff's left. Is Mike on Ed's left?
**Target:** I do not know

**Instruction:** Jake is ten feet away from me. Brynn is one hundred feet from Jake. Am I closer to Jake or Brynn?
**Target:** Jake

Table 4: Tasks on which Cappy's score shows obvious disadvantage compared with FLAN-T5-XXL's self-scoring.

connections. This might presents a considerable challenge to a model's "memory" capability. In the case of `simple_arithmetic_json` and `sufficient_information` tasks, the focus is on testing mathematical and commonsense logical abilities. The abilities of momorizing, math, and commonsense, have been demonstrated to be key advantages of LLMs [31, 9], and they are difficult to acquire through downstream training alone. Consequently, it is not surprising that the LLM's self-scoring outperforms Cappy's scoring in these tasks.

### 4.2.2 Performance Scaling with the Number of Samples

We study the relation between the adaptation performance and number of model generation samples. Specifically, we conduct this on a frozen FLAN-T5-XXL model, with three different numbers of samples, including 1 or 4 neucleus samples, or all the 17 samples as described in Section 4.2. Results are shown in Table 5, we can see that as the number of samples increases, Cappy consistently enhances task performance significantly, but the baseline Self-scoring doesn't provide significant performance boost increasing from 4 to 17 samples.

| Samples | 1 | 4 | 17 |
| --- | --- | --- | --- |
| Self-scoring | 26.90 | 31.15 | 32.02 |
| Cappy$_{\text{LARGE}}$(ours) | 26.90 | 33.64 | 36.93 |

Table 5: Performance of BIG-Bench adaptation on a frozen FLAN-T5-XXL with different numbers of samples.

### 4.2.3 Ablation Study

To verify the importance of the two key components in our proposed methodology, we carry out an ablation study, utilizing FLAN-T5-XL and -XXL's adaptation on BIG-Bench tasks. Specifically, in the downstream adapation, we ablate the pretraining of Cappy

| | Frozen FLAN-T5 | |
| --- | --- | --- |
| | XL | XXL |
| Cappy$_{\text{LARGE}}$ | 33.38 | 36.56 |
|   - w/o Cappy pretraining | 32.03 (-1.35) | 35.01 (-1.55) |
|   - w/o Data augmentation using LLM | 28.66 (-4.72) | 32.88 (-3.67) |

Table 6: Performance of BIG-Bench adaptation, before and after the ablation of Cappy's pretraining and data augmentation using LLM, numbers in the brackets are the performance drop.

that we described in Sec. 3.2, by using RoBERTa as the model initialization instead of Cappy. We also ablate the data augmentation using LLM, described in Sec. 3.2 during the downstream regression data construction for Cappy.

Table 6 shows the results. The ablation of either pretraining or data augmentation using LLM results in a noticeable decline in performance, thereby highlighting the significance of both these components within our proposed approach. Further, the performance decline scale reveals that the impact of data augmentation is more significant than the pretraining in the downstream adaptation of LLM.

# 5    Conclusion and Discussions

We deliver a lightweight pretrained scorer, *Cappy*, to enhance the performance and efficiency of multi-task LLMs. Cappy takes an instruction and a candidate response as input, and produces a score between 0 and 1. The score indicates an estimated correctness of the response with regard to the instruction. A weakly-supervised approach is proposed for the construction of Cappy's pretraining data in regression style. We suggest a candidate selection manner to apply Cappy into practical task-solving. Specifically, Cappy can be utilized either independently or in collaboration with an existing multi-task LLM, serving as an auxiliary component. Our experiments demonstrate that Cappy outperforms much larger multi-task LLMs, on 11 language understanding tasks. Besides, Cappy boosts FLAN-T5's performance on the adaptation to 45 complex tasks drawn from BIG-Bench, without requiring to finetune the LLM. Moreover, Cappy can effectively collaborate with other LLM adaptation strategies such as finetuning and in-context learning, providing further performance enhancement.

## Limitations and Future Directions

In Section 4.2.1, we have analyzed certain limitations of Cappy, specifically in the realm of mathematics and complex logical problems. Here, We detail some other limitations and future directions.

**Rouge-L Score for Weak Supervision**    In the construction of Cappy's pretraining data, Rouge-L serves as the metric to evaluate the correctness of model generations. However, Rouge-L may not be the optimal proxy for correctness of model generations, and there is no consensus across the ML and NLP community on the best metric across all the tasks. Presently, Rouge-L is commonly used in multi-task scenarios to report model performance for generation-style tasks [10]. Although Cappy's performance in our experiments demonstrate Rouge-L to be a reasonable design choice, investigating the most suitable metric for multi-task applications remains a highly valuable research direction.

**Answer Aggregation Across Multiple Generations**    The primary contribution of this work is the development and application of the pretrained model, Cappy, for multi-task applications. For sample selection from multiple model generations, we use a straightforward argmax manner that picks the sample with the largest score. However, recent research with nicely designed answer aggregation techniques [28] suggests potential avenues for refining the answer aggregation with Cappy, to further improve the performance of multi-task learning.

**Not Handling Tasks Outside the LLM's Expertise**    The aim of Cappy is to enhance the performance of tasks where the backbone LLM has a fundamental understanding of data inputs. However, Cappy doesn't impact the intrinsic ability of the LLM. It is also worth noticing that many multi-task LLMs, such as FLAN-T5 used in our experiments, already exhibit proficiency across a wide range of domains, encompassing areas like medicine, law, and coding.

**Single LLM Adaptation**    In the experiments of this work, we adapt a single LLM to several domains with Cappy. In the future, Cappy as a pretrained model can potentially be used in other creative ways beyond on single LLMs. For example, Cappy may work as a filter for generations from multiple LLMs. In this case, Cappy plays a role that selects the best LLM regarding a specific input.

**Resolving More Human-like Instructions and Leveraging Human Feedback**    In this work, our focus is multi-task learning where tasks are well-defined. In the future, Cappy can be potentially applied to resolve more "human-like" instructions where the tasks are often vaguely defined. To this end, it would be highly beneficial to leverage costly but high-quality human feedback data, which would require further algorithmic design. This is worth our further exploration.

## Acknowledgements

We thank Google Research for supporting Bowen Tan working as a student researcher in an internship. Eric Xing and Bowen Tan has also been graciously supported by NGA HM04762010002, NSF IIS1955532, NSF CNS2008248, NIGMS R01GM140467, NSF IIS2123952, NSF BCS2040381, an Amazon Research Award, NSF IIS2311990, and DARPA ECOLE HR00112390063. Zhiting Hu is partically supported by DARPA ECOLE HR00112390063.

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

# A   Comparison of Cappy with other adaptation methods

Cappy doesn't mean to beat other adaptation methods such as finetuning, in-context learning, and prompt tuning. Compared with these approaches, adaptation with Cappy is an alternative that is free from the constraints associated with storage, device memory, model accessibility, and training sample limitations. Moreover, Cappy makes no assumption about the backbone model, enabling seamless integration with other adaptations and providing steady and significant performance promotion at little additional cost. To illustrate this, we include an experiment below that combines Cappy with in-context learning and prompt-tuning, respectively.

Specifically, we add more comparison with prompt tuning and in-context learning in our BIG-Bench adaptation experiment with FLAN-T5-Large as the backbone model. For prompt tuning, we apply prefix tuning, which is usually considered suitable for generation tasks, with 20 virtual tokens. As demonstrated by the results presented in Table 7, Cappy offers further performance boost on top of both in-context learning and prompt tuning.

| Setting | Rouge-L |
|---|---|
| Frozen FLAN-T5-Large + Cappy$_{\mathrm{LARGE}}$(ours) | 30.75 |
| In-context learning + Nucleus | 23.65 |
| In-context learning + Self-scoring | 27.00 |
| In-context learning + Cappy$_{\mathrm{LARGE}}$(ours) | **31.84** |
| Prompt-tuning + Nucleus | 34.00 |
| Prompt-tuning + Self-scoring | 38.43 |
| Prompt-tuning + Cappy$_{\mathrm{LARGE}}$(ours) | **42.71** |

Table 7: Big-Bench performance under multiple adaptation settings with FLAN-T5-Large

