# OpenReview forum: "Cappy: Outperforming and Boosting Large Multi-Task LMs with a Small Scorer"
_NeurIPS.cc/2023/Conference — NeurIPS 2023 poster_

### Official Review · Reviewer_K4oy · 2023-06-25

**Soundness:** 3 good
**Presentation:** 3 good
**Contribution:** 2 fair
**Rating:** 6
**Confidence:** 4

**Summary:**

The authors propose training a small(er) classifier model (‘Cappy’) to predict an answer given a set of possibilities (if the answer set is closed) or multiple generations from an LLM (in the case of open-ended/generative tasks). The model is trained on data from T0, using LLMs to generate partially correct responses alongside positive and negative examples. They evaluate performance on T0 evaluation tasks (classification) and big-bench (generative) and find that Cappy outperforms same-size models, and consistently provides improvements over a base model when picking from generations.

**Strengths:**

The proposed approach is straightforward, using a trained classifier as a reranker for model generations. The improvements seem significant, and the fact that a classifier outperforms a similar-size generative model is interesting to see.

The data gathering scheme for Cappy’s training data is interesting, and using rouge-l as a proxy for gold labels is interesting and seems to work well in ablations.

The writing is clear and the paper is easy to follow.

**Weaknesses:**

**Comparisons to Best-of-N** - I’m somewhat concerned about the novelty of this work. The proposed method is very similar to the best-of-N sampling originally proposed in [1] (“Best-of-N sampling”) and discussed (sometimes under ‘reranking’) in [2] [3], [9], inter alia. These methods work by sampling multiple LM generations and then picking the answer with the highest reward as chosen by a trained reward model.
The three core differences between these prior works and Cappy I can detect are:
1. Different use in classification - Cappy directly uses the set of possible answers for classification tasks, while these prior works do not.
2. Training data - Cappy is trained on prompt source-based data with augmentation, which is interesting and seems to help performance, while (afaik) best-of-n uses trained reward models, which are often trained using preference datasets.
3. Application - best-of-n approaches are often applied and focussed on how they aid with aligning to human preferences, while the focus in this work is on improving over benchmarks.

While these are interesting differences, the authors do not compare to these prior works and explicitly make these comparisons. At the very least, I would expect these methods to be discussed in a related work section, and the decisions made in training Cappy contrasted to them. Ideally, they would be integrated as baselines/ablations to justify differences, highlighting the merits of Cappy over these approaches and covering what is needed to adapt these approaches to Cappy’s chosen setting.

**Missing prior work for reranking** - Cappy effectively is serving as a reranker in the generative case. There is a long history of work in answer reranking, that would be useful to discuss as related work (e.g., [4] for multi-choice qa, [5,7] for summarisation, [5,6] for open-domain qa).

**Further baselines and ablations** - There are other methods for training and applying rerankers, including contrastive training (as used to train a reranker in [7], with rouge as a proxy for human judgements), or encoding multiple samples at once as opposed to one-at-a-time (as done in [4]). It would be useful to test these changes, especially given the authors suggest that making use of contrastive information is helpful for Cappy in line 202. It would also be interesting to compare against self-consistency [8], another approach for improving models that makes use of multiple generations.

Given all of these things, I am inclined to recommend rejection for this paper in its current state. Further changes differentiating this approach against best-of-n sampling and further experiments exploring the full space of answer reranking would greatly improve this work, and may sway my opinion, but require significant extra work. I think the overall idea is interesting and evidently effective, but requires further work to make a substantial contribution. I hope the authors make these changes and are successful in the future!

[1] Stiennon et al. (2020). Learning to summarize from human feedback. NeurIPS. https://arxiv.org/pdf/2009.01325.pdf

[2] Bakker et al. (2022). Fine-tuning language models to find agreement among humans with diverse preferences. ArXiv. https://arxiv.org/pdf/2211.15006.pdf

[3] Glaese et al. (2022). Improving alignment of dialogue agents via targeted human judgements. ArXiv. https://arxiv.org/abs/2209.14375

[4] Kratzwald et al. (2019). RankQA: Neural Question Answering with Answer Re-Ranking. ACL. http://aclanthology.lst.uni-saarland.de/P19-1611.pdf

[5] Revaut et al. (2022). SummaReranker: A Multi-Task Mixture-of-Experts Re-ranking Framework for Abstractive Summarization. ACL. https://aclanthology.org/2022.acl-long.309.pdf

[6] Lee et al. (2018). Ranking Paragraphs for Improving Answer Recall in Open-Domain Question Answering. EMNLP. https://aclanthology.org/D18-1053.pdf

[7] Liu et al. (2021) SimCLS: A Simple Framework for Contrastive Learning of Abstractive Summarization. ACL. https://aclanthology.org/2021.acl-short.135/

[8] Wang et al. (2023). Self-Consistency Improves Chain of Thought Reasoning in Language Models. ICLR. https://arxiv.org/pdf/2203.11171.pdf

[9] Dubois et al. (2023). AlpacaFarm: A Simulation Framework for Methods that Learn from Human Feedback. ArXiv. https://arxiv.org/abs/2305.14387

Edit: The authors have responded with some of the experiments and comparisons I asked for, with positive results, and so I'm happy to raise my score.

**Questions:**

1. How is cappy different to best-of-n techniques proposed in the papers mentioned above? Do you think using preference data already available could help or augment your data collection strategy?
2. Did you explore contrastive objectives in pretraining cappy? How about encoding multiple answers at once?
3. Do you have further details on the distribution of rouge-l scores you used in the data augmentation setup? It would be interesting to see this, and do you think there are any limitations in using rouge-l against something like cosine similarity for this?

**Limitations:**

I think the authors effectively discuss limitations. They miss some elements such as Cappy requiring multiple generations (thus increasing its inference cost over using a single-generation) and complicating the model pipeline, but cover Cappy’s weakness in the realm of complex logical problem-solving / mathematics, and its current reliance on supervised datasets.

---

> ### Author Rebuttal · Authors · 2023-08-10
>
> Thank you for your feedback that our data construction method is interesting, that Cappy’s improvements over other methods are significant, and that our writing is clear and easy to follow.
>
>
> **Comparison with Reranking and best-of-N sampling**\
> We appreciate the suggestions on various sample selection methodologies and your insights on the distinctions between Cappy and these techniques.
>
> To summarize and extend your valuable observations, the key difference between Cappy and previous reranking and best-of-N sampling methods lies in two aspects: (1) Compared with methods that conduct ranking based on reward models [1, 2, 3, 9], Cappy doesn’t rely on expensive human-annotated data, which enables our large-scale pretraining; (2) Unlike reranking methods specifically tailored for QA [4, 6] or summarization [5, 7], Cappy exhibits broader generalizability across multi-task settings.
>
> However, we would also like to clarify that our key contribution is constructing a large-scale dataset, delivering the pretrained model Cappy, and applying Cappy into multi-task applications. In terms of sample selection, we actually use a rather trivial argmax manner that picks the sample with the largest score. This sample selection is also used in our Self-scoring baseline. As of now, we have not incorporated any fancy sample selection techniques such as best-of-N rejection sampling [1] or answer aggregation [8]. That being said, we will definitely explore more suitable sample selection strategies for Cappy in our future research.
>
>
>
> **Contrastive objective, encoding multiple answers at once, human preference data**\
> We do incorporate contrastive information during the training of our model. However, this is not achieved through an explicit contrastive loss function or a modeling to encoding multiple answers at once. Instead, the contrastive information is sourced from our pretrain data, where there are several examples sharing the same instruction but have different responses and score annotations. We have demonstrated the effectiveness of this incorporation, by Cappy’s performance in the multi-task scenarios presented in our experiments.
>
> As for the use of human preference data, we acknowledge this can be very beneficial. However, integrating human-annotated data would require further algorithmic design. It's also worth noticing that obtaining human preference data can be costly, and the volume might be insufficient for large-scale pretraining. We remain committed to exploring strategies for effectively integrating human feedback in our future work.
>
>
> **Using Rouge-L score as a weak supervision**\
> Thanks for the suggestion to analyze the Rouge-L distribution. Among all of our pretrain data with 160M data points, there are 30M samples are labeled 1.0, 90M samples labeled 0.0, and the other 40M has score annotations between 0 to 1. Specifically, the average of the third part is 0.3 and the whole distribution can be visualized in our author rebuttal pdf. From this analysis, we can see that all the scores ranging from 0 to 1 have significant number of samples, which demonstrates the effectiveness of our data construction.
>
> We agree that Rouge-L may not be the optimal proxy of correctness of model generations. Cosine similarity may measure the semantic information better than the n-gram based Rouge-L. However, using Rouge-L could be a more suitable design choice for our large-scale pretraining, because (1) cosine similarity is less efficient: it would be time-consuming to get an embedding and run a similarity score for every example in the large pretrain dataset. (2) besides, it would require specific embedding models for tasks in a specific domain, while Rouge-L doesn’t have this limitation. (3) The community has not reached a consensus on the best metric across all the tasks. Nonetheless, Rouge-L is commonly used in multi-task scenarios to report model performance for generation-style tasks, such as in OPT-IML paper. (4) As a weak supervision for a subset of data in our large-scale pretraining, we acknowledge and accept that some data examples might not be perfectly labeled. (5) Moreover, Cappy’s performance in our experiments is further evidence for Rouge-L to be a reasonable design choice.
>
> Investigating the most suitable metric for multi-tasking is a highly valuable research direction. Thank you for pointing this out, and we will keep exploring this in the future.
>
>
> Thanks for suggesting more potential limitations of our work. We will add our discussion to all these points in our next paper update.

---

> > ### Comment · Reviewer_K4oy · 2023-08-15
> > **Re: Rebuttal**
> >
> > Hi, thank you for the detailed response! I agree that the collection/generation of the dataset for training the cappy model is novel and interesting, and the way cappy is applied is simple. Additionally, thank you for the clarification with the contrastive data, that makes sense, and thank you for the rouge-l data. It’s interesting it's mostly centred around .2, which suggests lots of generations with low overlap - it makes me wonder if better balancing the distribution might improve results somewhat (but this is beyond the scope of this paper).
> >
> > However, I disagree that the methods in [1,8] are significantly different or fancier than what is proposed here. As stated on page 24 of [1], their best-of-n procedure is to “Sample N summaries from a supervised baseline at temperature 0.7, score them with a reward model, and take the summary with the highest score”. This is very similar to Cappy (argmax based on an answer rating module), but using slightly different decoding strategies, and using a reward model instead of Cappy. There are publicly available datasets for training reward models (e.g. https://huggingface.co/datasets/stanfordnlp/SHP, https://huggingface.co/datasets/Anthropic/hh-rlhf), and I think it is reasonable to compare a model trained using the Cappy data to models trained using these datasets - my guess is cappy would do better since it is focussed on correctness instead of helpfulness/harmfulness, but since the method is so close to prior best-of-n work, I think it is a necessary comparison.
> >
> > In my personal opinion, this is why I lean still reject and am not updating my score, but if other reviewers and AC are satisfied the method is novel enough, and the comparisons are rigorous enough, I think the other aspects of the work are solid and the empirical results certainly still useful for the community.

---

> > > ### Author Response · Authors · 2023-08-20
> > > **Re: Re: Rebuttal**
> > >
> > > Thank you for acknowledging that our response has addressed most of your previous concerns regarding our novelty, the incorporation of contrastive information, and the usefulness of our empirical results.
> > >
> > > We would like to emphasize that the focus of our work is multi-task learning where tasks are clearly defined, and actually the incorporation of human feedback is largely orthogonal to this work. Specifically, our application domain is consistent with those well-accepted pretrained multi-task LLMs mentioned in our paper, such as T0, FLAN, and OPT-IML. Notably, all of these models do not rely on costly human annotations during pretraining, and they often would not be directly compared with models using human preference data, either in multi-task learning or RLHF literatures, mainly based on the consideration of (1) fair comparison, and (2) their different application domains. Nonetheless, we appreciate your suggested comparison mentioned in page 24 of [1], i.e., using the reward model trained by human preference data to conduct argmax sample selection without applying RL algorithms. We include an experiment about it in the below.
> > >
> > > Specifically, we add two baselines with publicly available reward models trained by LAION-AI (OpenAssist), including
> > >
> > >
> > > * RLHF RM-large: https://huggingface.co/OpenAssistant/reward-model-deberta-v3-large-v2
> > > * RLHF RM-base: https://huggingface.co/OpenAssistant/reward-model-deberta-v3-base
> > >
> > > They are trained on the combination of these four human preference datasets:
> > > * Summarize_from_feedback (the human preference dataset in [1]): https://huggingface.co/datasets/openai/summarize_from_feedback
> > > * Anthropic_hh-rlhf (the one in your response): https://huggingface.co/datasets/Anthropic/hh-rlhf
> > > * Webgpt_comparisons: https://huggingface.co/datasets/openai/webgpt_comparisons
> > > * Synthetic-instruct-gptj-pairwise: https://huggingface.co/datasets/Dahoas/synthetic-instruct-gptj-pairwise
> > >
> > > As a result, our Table 1 is updated as below. Consistent with our initial expectations, Cappy outperforms the baselines with reward models of RLHF. It is also interesting that the reward models trained from human preference data can outperform some multi-task LLMs like OPT-IML-30B. That also reflects the advantage of incorporating contrastive information against relying exclusively on ground truth data. We will update the results and add the discussion in our next paper update.
> > >
> > > | Model                | Accuracy |
> > > |----------------------|----------|
> > > | BART0-base (140M)    | 45.7     |
> > > | BART0-large (400M)   | 50.2     |
> > > | OPT-30B              | 47.6     |
> > > | OPT-IML-30B          | 51.3     |
> > > | OPT-175B             | 49.3     |
> > > | OPT-IML-175B         | 56.5     |
> > > | T0-11B               | 58.2     |
> > > | **RLHF RM-base (185M)**  | 43.6     |
> > > | **RLHF RM-large (435M)** | 53.3     |
> > > | Cappy-base (120M)    | 49.9     |
> > > | Cappy-large (360M)   | 56.6     |

---

> > > > ### Comment · Reviewer_K4oy · 2023-08-21
> > > > **Re: Re: Re: Rebuttal**
> > > >
> > > > Thank you for the response! This experiment is very solid and I'm happy to raise my score with it being added to the paper. It would also be interesting to see how the reward models perform in the generative setting, but it's already very interesting that they outperform some OPT models.

---

### Official Review · Reviewer_hbCr · 2023-07-03

**Soundness:** 3 good
**Presentation:** 2 fair
**Contribution:** 2 fair
**Rating:** 6
**Confidence:** 4

**Summary:**

This paper introduces an auxiliary module called Cappy, which aims to enhance the performance of large language models. Cappy operates by taking a task instruction and a proposed response as input, and it estimates the quality score for the response. The training process involves creating a dataset that combines correct and mismatched answers with various instructions, and the Cappy scorer is then trained using a regression objective. In downstream adaptation, the small scorer, Cappy, is fine-tuned on a small dataset that follows the same approach as the training set. During testing, predictions are determined by selecting the answer choice with the highest score. In essence, Cappy acts as a valuable filter that enhances the overall quality of generated responses.

**Strengths:**

The idea of leveraging a lightweight scorer as a ‘’filter’’ is interesting. Fine-tuning the filters on downstream tasks can be efficient and practical.

**Weaknesses:**

Cappy itself is only a filter for in-domain responses and can not impact the intrinsic ability of a pre-trained LLM. This method cannot handle tasks outside the LLM's expertise, emphasizing the need for a sufficiently general and powerful LLM. Alternatively, fine-tuning LLMs, as shown in Table 3, may be necessary but costly. By comparing able 2 and Table 3, fine-tuning is still very essential for task adaptation.

**Questions:**

Please refer to the weaknesses.

**Limitations:**

No additional limitations.

---

> ### Author Rebuttal · Authors · 2023-08-10
>
> Thank you for your positive comments that our idea is interesting, and our proposed downstream adaptation approach is efficient and practical.
>
> **Not handling tasks outside the LLM's expertise**\
> Indeed, the primary objective of Cappy is to enhance performance of tasks where the backbone LLM possesses a fundamental understanding of the data input. Notably, many multi-task LLMs, including FLAN-T5 utilized in our experiments, exhibit proficiency across a wide range of domains, encompassing areas like medicine, law, and coding.
>
>
> **Comparison of Cappy with finetuning**\
> We would like to clarify that Cappy doesn’t mean to beat other adaptation methods, especially finetuning. Compared with other adaptation approaches, Cappy is an alternative free from the constraints associated with storage, device memory, model accessibility, and training sample limitations.
>
> Moreover, Cappy doesn’t have any assumption about the backbone model, enabling seamless integration with other adaptations. As we show in Table 3, Cappy provides steady and significant performance promotion with little additional cost,

---

> > ### Comment · Reviewer_hbCr · 2023-08-16
> >
> > I appreciate your response. The strength of Cappy seems to lie in its ability to **adapt a single LLM to several domains** using lightweight filters, which is difficult to achieve through fine-tuning. I think emphasizing this in the paper could enhance its presentation.
> >
> > However, this is a good and practical work. I will keep my initial rating.

---

> > > ### Author Response · Authors · 2023-08-20
> > >
> > > Thanks for your feedback that this is a good and practical work!
> > >
> > > Thanks for the suggestion regarding our presentation. We agree that our adaptation experiments can be summarized as “adapting a single LLM to several domains using a lightweight filter”. And we will definitely make this point clearer in the revised version.
> > >
> > > To add a little bit, in the future, Cappy as a pretrained model can potentially be used in other creative ways beyond on single LLMs. For example, Cappy might be used as a filter for generations from multiple LLMs. In this case, Cappy plays a role that selects the best LLM regarding a specific input. We will add the discussion, potentially with related experiments, to the next version of our paper.

---

### Official Review · Reviewer_XRdH · 2023-07-03

**Soundness:** 3 good
**Presentation:** 2 fair
**Contribution:** 3 good
**Rating:** 6
**Confidence:** 4

**Summary:**

Adapting SotA LLMs to novel tasks is difficult given their extreme size, yet is generally more effective than in-context learning alone. To address this, the authors propose “Cappy”, a scoring model (leveraging RoBERTa as a backbone) which is trained to score pairs of (instruction, response) for any task instruction and corresponding response. Cappy is pre-trained on many examples of ((instruction, response) -> score); after pre-training, Cappy can be easily fine-tuned (because of it’s small size) to a given task and then used as an augmentation to LLM generation by acting as a ranking mechanism atop several LLM generated responses.

To create pre-training data for Cappy, the authors use the PromptSource dataset, training the model to predict 1 for correct pairs of (prompt, response) and 0 for mismatched pairs. Additionally, the authors generate additional responses using a large LLM (e.g. FLAN) and use the Rouge-L between the generated response and gold response as the regression target for the pair (prompt, generated response). To test the validity of Cappy, the authors evaluate Cappy in isolation, with no additional fine-tuning, on 11 held-out classification tasks from PromptSource. They show that Cappy consistently ranks the correct answer higher than many instruction-tuned LLMs, including OPT-IML-175B.

Next, the authors evaluate Cappy on LLM augmentation using the BIG-Bench benchmark. For each task, Cappy is first fine-tuned, then used as a scoring function for multiple generations from a frozen FLAN-T5 LLM. Cappy consistently outperforms the proposed sampling methods, most notably improving upon the LLMs own self-scoring and In-Context Learning. Moreover, the authors show that Cappy can improve upon a FLAN-T5 model’s performance even when the LLM parameters are fine-tuned as well.

Finally, the authors present an ablation study of their proposed training strategy for Cappy, showing that both the overall pre-training strategy and using Rouge-L regression targets from LLM generations are important to Cappy’s success.

**Strengths:**

- Cappy represents a light-weight option to adapt LLMs to a given target task which requires no updating of parameters to the LLM nor any back-propagation through the LLM whatsoever (e.g. this is not true for adapter layers or prompt tuning).
- Moreover, even without adapting to a given task, Cappy presents some benefits over generic LLM sampling methods for task-specific predictions.
-  Additionally, Cappy can ostensibly be added on top of any LLM without additional training (i.e. train once, use anywhere).
- Cappy is a very simple idea, and is easy to understand, yet the usage of a scorer in this capacity, as well as the training and data augmentation scheme, is novel to the best of my knowledge.
- The ablation study is helpful in demonstrating the benefits of the proposed pre-training and data-augmentation scheme.

**Weaknesses:**

- Cappy is not compared to other parameter-efficient adaptation methods. While there is some justification given, because other methods require storing forward activations and back-propagating through LLM parameters, it nevertheless makes judging the effectiveness of Cappy difficult.
- Moreover, Cappy’s improvements over a frozen LLM are comparatively small, e.g. while Cappy improves Flan-T5 Large performance by ~7%, fine-tuning alone increases performance by ~22%. It’s therefore unclear how significant Cappy’s benefit is without comparison to other adaptation methods.
- In the zero-shot experiments (4.1) it is claimed that the improved performance of Cappy over much larger LLMs can be attributed to Cappy’s scoring-based learning strategy. However, T0 outperforms Cappy and T0 is significantly smaller than the OPT models; moreover, T0 is also trained on PromptSource only, similar to Cappy, and unlike the other LLMs. Thus, it seems that at least some of Cappy’s performance benefits compared to other LLMs may arise from domain mismatch between training and test domains, rather than the proposed scoring function.
- While Cappy does only require a small number of parameters to train, inference requires a number of decoder forward passes to generate multiple candidate responses from an LLM (it's not clear how Cappy scales with the number of samples provided, also); to my knowledge, other adaptation methods such as Adapter Layers do not have the same limitation. So while it saves computation on one end, it does slightly increase computation on another end, and exploring this trade-off would be useful.

**Questions:**

- I think it would be really helpful to understand how Cappy compares to other parameter efficient adaptation methods. Do you have any experiments that indicate that Cappy is e.g. comparable to prompt-tuning in terms of it's benefits?
- Can you provide an analysis of how much additional computation is required to sample many generations for Cappy to score?
- Also, do you know how Cappy's performance scales with the number of generated samples? e.g. does it's performance flatten out after only 4 samples, or can it's performance increase significantly if several generations are considered?

**Limitations:**

Yes.

---

> ### Author Rebuttal · Authors · 2023-08-10
>
> Thank you for your supportive comments, including that our approach is novel, our delivered model Cappy is light-weight and beneficial, and our ablation study is well conducted.
>
> **Comparison of Cappy with other adaptation methods**\
> We would like to clarify that Cappy doesn’t mean to beat other adaptation methods such as finetuning, in-context learning, and prompt tuning. Compared with these approaches, adaptation with Cappy is an alternative free from the constraints associated with storage, device memory, model accessibility, and training sample limitations. Moreover, Cappy doesn’t have any assumption about the backbone model, enabling seamless integration with other adaptations, where Cappy also provides steady and significant performance promotion with little additional cost. To illustrate this, we add an experiment below that combines Cappy with in-context learning and prompt-tuning.
>
> Specifically, we add more comparison with prompt tuning and in-context learning in our BIG-Bench adaptation experiment with FLAN-T5-Large as the backbone model. For prompt tuning, we apply prefix tuning, which is usually considered suitable for generation tasks, with 20 virtual tokens. As demonstrated by the results presented below, Cappy offers further performance boost on top of in-context learning and prompt tuning.
>
> | Setting                                   | Rouge-L |
> |-------------------------------------------|---------|
> | frozen FLAN-T5-Large + Cappy-Large (ours) | 30.08   |
> | In-context learning + Nucleus             | 22.59   |
> | In-context learning + Self-scoring        | 27.00   |
> | In-context learning + Cappy-Large (ours)  | **31.84**   |
> | Prompt-tuning + Nucleus                   | 34.00   |
> | Prompt-tuning + Self-scoring              | 38.43   |
> | Prompt-tuning + Cappy-Large (ours)        | **42.71**   |
>
>
> **Train/test domain mismatch**\
> Actually, PromptSource is part of OPT-IML pretrain data, and OPT-IML encompasses even more training tasks compared with Cappy, so that the domain mismatch might not be a benefit of Cappy. Thanks for asking this! We will make this more clear in our next paper update.
>
> **Computational cost of multiple generations**\
> Given Cappy’s small size, the predominant computational overhead arises from generating multiple samples using the backbone LLM. That is, if N samples are generated, the computational effort would be N times that of a standard single-sample generation, assuming the samples are not batch-processed.
>
> However, collecting multiple samples for a single prediction is a common operation in many algorithms, such as self-consistency [1], and some re-ranking techniques designed for QA [2] and summarization [3]. Furthermore, by leveraging the batch processing capabilities of GPUs/TPUs, the actual time overhead, when compared with single-sample generation, can be significantly less than N-fold. This efficiency potentially explains the continued popularity of these techniques.
>
>
> [1] Self-Consistency Improves Chain of Thought Reasoning in Language Models\
> [2] RankQA: Neural Question Answering with Answer Re-Ranking\
> [3] SummaReranker: A Multi-Task Mixture-of-Experts Re-ranking Framework for Abstractive Summarization
>
>
>
> **How performance scales with the number of samples**
>
> We appreciate such a meaningful suggestion. We conduct this on a frozen FLAN-T5-11B model, with three settings as below:
> * 1 sample:    a single nucleus sample (top-p=0.95)
> * 4 samples:  4 nucleus samples
> * 20 samples: 4 samples from 5 different decoding methods (Random Sampling, Temperature, Top-K, Nucleus, Beam Search).
>
> Results are shown as below, as the number of samples increases, Cappy consistently enhances task performance significantly, in contrast with the baseline Self-scoring.
>
> |                    | 1 sample | 4 samples | 20 samples |
> |--------------------|----------|-----------|------------|
> | Self-scoring       | 27.33    | 31.15     | 32.62      |
> | Cappy-Large (ours) | 27.33    | **33.64**     | **36.56**      |
>
>
> Thanks for the suggestions and meaningful points above! We will add all the discussion above to the next version of our paper.

---

### Official Review · Reviewer_uzD3 · 2023-07-07

**Soundness:** 3 good
**Presentation:** 4 excellent
**Contribution:** 3 good
**Rating:** 7
**Confidence:** 3

**Summary:**

In this paper, the authors tackle the challenge of computational requirements and memory constraints in fine-tuning Large Language Models (LLMs) by introducing an innovative approach that enhances LLM performance without the need for backpropagation through the LLM or access to its parameters. They propose Cappy, a pre-trained scorer that evaluates texts generated by LLMs based on downstream instructions and assigns them a score ranging from 0 to 1.

To create Cappy, the authors curate a diverse collection of 39 samples from the PromptSource dataset, including good (ground truth), bad (random), and intermediate examples. The intermediate examples are generated using top-k and top-p sampling from BART0 and T0-3B models. Cappy assigns a score to each example based on the ROUGE-L metric compared to the ground truth response. This dataset is used to create a regression task and train a RoBERTa model that serves as the foundation for Cappy.

The authors demonstrate improved accuracy across 11 classification held-out tasks from PromptSource compared to OPT, OPT-IML, T0, and BART0. They also showcase Cappy's effectiveness as an LLM booster for downstream adaptation on the BIG-Bench generative tasks, outperforming other selection strategies and achieving higher ROUGE-L scores on both frozen and fine-tuned FLAN-T5 models.

Additionally, the paper discusses scenarios where Cappy performs less favorably than the "self-scoring" strategy on certain BIG-Bench tasks, suggesting that the lack of "memory" may contribute to this performance difference. Furthermore, the authors conduct an ablation study, revealing that data augmentation using LLMs is more crucial for improved performance than pre-training Cappy, although pre-training still offers benefits.

Overall, the paper presents an innovative approach, Cappy, for boosting LLM performance without backpropagation or parameter access. The experiments showcase its superiority in classification tasks and downstream adaptation, while highlighting the importance of data augmentation and the potential limitations related to memory.

**Strengths:**

- The idea of an auxiliary performance booster is an intriguing concept. The demonstrated improvements over the selected baselines highlight the effectiveness of this approach in enhancing language model performance.
- The utilization of LLM generations for data augmentation and the regression-based evaluation using ROUGE-L scores is a novel and innovative methodology employed in this work.
- Cappy introduces a unique integration of samples from LLMs and ground-truth information, leveraging correctness scores to enhance the quality of text generations. This ability to distinguish between good and poor responses is a novel idea that has the potential to enhance the performance of all language models.
- Despite having significantly fewer parameters compared to large language models used in zero-shot baselines, pre-trained Cappy achieves comparable performance. This contribution is valuable as it demonstrates that Cappy can achieve performance on par with larger models, offering a more efficient and resource-friendly solution.
- The versatility of Cappy is showcased by its potential for downstream adaptation and further fine-tuning to improve task-specific performance. This adaptability highlights the broad applicability and utility of Cappy as a solution in various contexts.

**Weaknesses:**

- The performance of Cappy, trained on BART0 and T0 generations, does not consistently outperform T0 and OPT-IML. While the zero-shot performance is better than BART0 (a relatively smaller model), it falls short compared to T0 in most cases. The evidence supporting the claim of boosting language model performance is not sufficiently strong, and the improvements are not consistently observed.
- The training of Cappy to optimize ROUGE-L scores against ground truth data raises concerns about the choice of this metric. ROUGE-L may not necessarily be the most optimal metric for evaluation, as alternative responses with low ROUGE-L scores may be valid and informative. Including additional metrics for evaluation would provide a more comprehensive understanding of the overall increase in performance.
- The discussion on the tasks in BIG-Bench where Cappy performs worse than the "self-scoring" strategy is insufficient. Given the expectations for Cappy to perform at least as well, if not better, further analysis and exploration of these instances would strengthen the paper and provide a clearer understanding of the limitations of Cappy's performance.

**Questions:**

- Could you provide an example illustrating how the limitation of instruction lengths affects in-context learning? Adding a specific example would enhance the understanding of this limitation and its implications.
- In the context of downstream adaptation, it is not entirely clear whether the same LLM that is being augmented is used to generate the synthetic data. Clarifying this aspect in the paper would help readers better comprehend the process of downstream adaptation and the relationship between the augmented data and the LLM.
- The reasoning behind the requirement for more "memory" in tasks like "sufficient_information" is unclear. It would be beneficial to provide a more detailed explanation either in the paper or in the appendix to shed light on the specific aspects that require increased memory in such tasks.
- The results show that removing the Cappy pre-training step does not significantly impact the downstream score on BIG-Bench, implying that fine-tuning with augmented data from LLMs is sufficient. Have you explored the performance of RoBERTa initialization with data augmentation using LLMs? It would be insightful to investigate and report the results of this comparison to further validate the efficacy of the proposed approach.

Minor Issues:
- Line 186: PromprSource -> PromptSource
- Line 285: 360 -> 360 M
- Line 222: freezed -> frozen - This issue exists at several places in the paper.

**Limitations:**

The authors thoroughly discuss various limitations, including mathematical complexities, reliance on supervised datasets, absence of multi-lingual extensions, and other important concerns. However, it would be beneficial to further address limitations related to the reliance on generations from existing LLMs and potential biases that may arise as a result. Additionally, the use of ROUGE-L as the sole metric for regression and evaluation may not provide a comprehensive and accurate portrayal of the overall performance. It would be valuable to acknowledge this limitation and discuss the potential impact on the interpretation and generalization of the results.

---

> ### Author Rebuttal · Authors · 2023-08-10
>
> Thank you for your supportive feedback that our idea of auxiliary performance booster is intriguing, our methodology is innovative, our delivered model Cappy is versatile, and our multi-task application with Cappy is valuable, efficient and resource-friendly.
>
> **Boosting language model performance** \
> Indeed, Cappy doesn’t beat T0 in the zero-shot setting. However, it actually gets an accuracy very close to that of T0 (Cappy 56.6 v.s. T0 58.2, as illustrated in Table 1), considering the substantial difference of their model sizes (Cappy 360M v.s. T0 11B). Furthermore, our BIG-Bench adaptation experiments demonstrate that Cappy provides steady performance improvement for LLMs of varied sizes, and either frozen or finetuned. Based on all these observations above, we overall conclude that Cappy boosts multi-task language models.
>
> **Using Rouge-L score as a weak supervision**\
> We agree that Rouge-L may not be the optimal proxy for correctness of model generations. Our choice of Rouge-L is primarily based on three considerations as below: (1) The community has not reached a consensus on the best metric across all the tasks. Nonetheless, Rouge-L is commonly used in multi-task scenarios to report model performance for generation-style tasks, such as in OPT-IML paper. (2) As a weak supervision for a subset of data in our large-scale pretraining, we acknowledge and accept that some data examples might not be perfectly labeled. (3) Moreover, Cappy’s performance in our experiments is further evidence for Rouge-L to be a reasonable design choice.
>
> That being said, investigating the most suitable metric for multi-task applications is a highly valuable area of research. Thank you for pointing this out, and we will keep exploring this in the future.
>
>
> **Further analysis on BIG-Bench experiments**\
> We appreciate the suggestion and we will deeply dive into more tasks on which Cappy doesn't perform better than Self-scoring.
>
> Here we present another two such tasks as below – “operators” and “physics_questions”. In line with the tasks we elaborated in our paper, they also demand heavy math and commonsense/physics knowledge, respectively.
>
> “operators”:\
> *Instruction: Given the definition of the op operator, compute the result. op n1 n2 ... nn extracts the last multiple of 8 from the n listed numbers. op 4 32 128 132 =*\
> *Target: 128*
>
> “physics_questions”\
> *Instruction: Q: The historic Stanley Center for the Arts in Utica, New York is the proud owner of the world’s largest LED chandelier. The chandelier is 35 feet wide, 17 feet tall and has a mass of 2900 kg. It is directly supported by four cables which make an angle of 63° with the horizontal. Determine the tension in the cables. A:*\
> *Target: 7974 N*
>
>
> **Response to Questions**
> * For example, consider an LLM with a large context length of 2048, and a task such as NLI, where the average length of each training example is approximately 32. In this case, in-context learning can accomondate about 2048 / 32 = 64 demonstrations within the input context at its maximum. However, many complex downstream tasks have tens of thousands of training examples, such as many tasks in BIG-Bench.
> * Yes, in the downstream adaptation, the augmented data for Cappy’s finetuning is from the model to be enhanced.
> * The task requiring memory ability is actually “codenames”, where a long list of words is given and the model needs to remember and identify a couple of them with a given association.
> * Our ablation of Cappy pretraining is already using RoBERTa initialization with data augmentation. We will make this clearer in our next paper update.
>
> Thanks for pointing out our typos and suggesting more potential limitations! We will fix/add them in the next version of our paper.

---

### Author Rebuttal · Authors · 2023-08-10

We would like to express our gratitude to all the reviewers for their insightful comments. We are encouraged by the reviewers' appreciation that our idea of auxiliary performance booster is intriguing (uzD3), that our methodology is novel and innovative (uzD3, XRdH), that our delivered model Cappy is valuable, efficient, resource-friendly, versatile and practical (uzD3, hbCr), that our ablation study is well conducted (XRdH, K4oy), and that our paper writing is clear and easy to follow (K4oy).

---

### Decision · Program_Chairs · 2023-09-21

**Decision:**

Accept (poster)

**Comment:**

The paper introduces a new method for improving instruction following in language models, by training a small scoring model to rate the quality of a language model generation. Rouge-L is used to automatically judge the quality of a large set of LM generations, and then Roberta is finetuned as a scoring model. Experiments demonstrate that a small ranking model can still lead to improvements on Big Bench. Reviewers appreciated that the technique is simple, effective and practical, and I recommend this paper for acceptance.